# *Staphylococcus aureus* Breast Implant Infection Isolates Display Recalcitrance To Antibiotic Pocket Irrigants

Jesus M. Duran Ramirez,[a,b] Jana Gomez,[a] Blake M. Hanson,[b,c] Taha Isa,[a] Terence M. Myckatyn,[d] Jennifer N. Walker[a,b*]

aDepartment of Microbiology and Molecular Genetics, McGovern Medical School, University of Texas Health Science Center, Houston, Texas, USA

bDepartment of Epidemiology, Human Genetics, and Environmental Sciences, Center for Infectious Diseases, School of Public Health, University of Texas Health Science Center, Houston, Texas, USA

cCenter for Antimicrobial Resistance and Microbial Genomics, McGovern Medical School, University of Texas Health Science Center, Houston, Texas, USA

dDivision of Plastic and Reconstructive Surgery, Washington University School of Medicine, St. Louis, Missouri, USA

**ABSTRACT**  Breast implant-associated infections (BIAIs) are the primary complication following placement of breast prostheses in breast cancer reconstruction. Given the prevalence of breast cancer, reconstructive failure due to infection results in significant patient distress and health care expenditures. Thus, effective BIAI prevention strategies are urgently needed. This study tests the efficacy of one infection prevention strategy: the use of a triple antibiotic pocket irrigant (TAPI) against *Staphylococcus aureus*, the most common cause of BIAIs. TAPI, which consists of 50,000 U bacitracin, 1 g cefazolin, and 80 mg gentamicin diluted in 500 mL of saline, is used to irrigate the breast implant pocket during surgery. We used *in vitro* and *in vivo* assays to test the efficacy of each antibiotic in TAPI, as well as TAPI at the concentration used during surgery. We found that planktonically grown *S. aureus* BIAI isolates displayed susceptibility to gentamicin, cefazolin, and TAPI. However, TAPI treatment enhanced biofilm formation of BIAI strains. Furthermore, we compared TAPI treatment of a *S. aureus* reference strain (JE2) to a BIAI isolate (117) in a mouse BIAI model. TAPI significantly reduced infection of JE2 at 1 and 7 days postinfection (dpi). In contrast, BIAI strain 117 displayed high bacterial burdens in tissues and implants, which persisted to 14 dpi despite TAPI treatment. Lastly, we demonstrated that TAPI was effective against *Pseudomonas aeruginosa* reference (PAO1) and BIAI strains *in vitro* and *in vivo*. Together, these data suggest that *S. aureus* BIAI strains employ unique mechanisms to resist antibiotic prophylaxis treatment and promote chronic infection.

**IMPORTANCE**  The incidence of breast implant associated infections (BIAIs) following reconstructive surgery postmastectomy remains high, despite the use of prophylactic antibiotic strategies. Thus, surgeons have begun using additional antibiotic-based prevention strategies, including triple antibiotic pocket irrigants (TAPIs). However, these strategies fail to reduce BIAI rates for these patients. To understand why these therapies fail, we assessed the antimicrobial resistance patterns of *Staphylococcus aureus* strains, the most common cause of BIAI, to the antibiotics in TAPI (bacitracin, cefazolin, and gentamicin). We found that while clinically relevant BIAI isolates were more susceptible to the individual antibiotics compared to a reference strain, TAPI was effective at killing all the strains *in vitro*. However, in a mouse model, the BIAI isolates displayed recalcitrance to TAPI, which contrasted with the reference strain, which was susceptible. These data suggest that strains causing BIAI may encode specific recalcitrance mechanisms not present within reference strains.

**KEYWORDS**  breast implant infections, *Pseudomonas aeruginosa*, *Staphylococcus aureus*, triple antibiotic pocket irrigant, antibiotic resistance

Address correspondence to Jennifer N. Walker, jennifer.n.walker@uth.tmc.edu.

*Present address: Jennifer N. Walker, University of Texas Health Science Center, Houston, Texas, USA.

The authors declare a conflict of interest. Myckatyn receives royalties for product development, funds for an investigator initiated trial associated with acellular dermal matrices in breast reconstruction, and advisory board remuneration from RTI Surgical. He receives an investigator initiated award from Sientra that studies the metabolomics of breast tissue expander infection. None relate directly to the topic matter of this study and no industry funds were received for completing this study. He has no other disclosures. All other authors declare no other conflicts of interest.

Nearly 300,000 breast prostheses are placed annually in the US for cosmetic and reconstructive purposes, of which 1% to 35% become infected (1–3). These infections can cause significant patient morbidity, as they can result in pain, fluid collections, fever, tissue necrosis, and device failure (1, 3–11). Notably, most breast implant-associated infections (BIAIs) occur during reconstructive breast implant-based surgery following mastectomy due to cancer, which have infection rates as high as 35% (1–3). For women with cancer, BIAI can lead to additional complications and delay of adjuvant therapies, such as chemotherapy and radiation. Furthermore, treatment of BIAIs require the explantation of the infected prosthesis, which necessitates additional surgical procedures and administration of broad-spectrum antibiotics, increasing the risk of disseminated infection and escalating health care costs (12, 13). Thus, BIAIs are a significant health burden for women, and prevention has become a priority (2, 5, 14–16).

Current evidence-based prophylaxis strategies involve the administration of preoperative intravenous antibiotics—typically first- or second-generation cephalosporins—and surgical skin scrubs at the incision site (17, 18). However, infection rates remain high. Thus, additional preventive strategies that involve the flushing of the surgical pocket with a triple antibiotic pocket irrigant (TAPI) have been implemented by surgeons in an effort to further reduce infection rates (17, 19). TAPIs typically consist of 50,000 U bacitracin, 1 g cefazolin, and 80 mg gentamicin diluted in 500 mL of saline and are introduced into the surgical pocket prior to breast prosthesis placement (15, 16). Recent *in vitro* studies suggest that prophylactic antibiotic irrigation strategies, such as TAPI, may be efficacious against some of the most common causes of BIAI, including *Staphylococcus aureus* and *Pseudomonas aeruginosa* (4, 6, 16, 20–23). While these studies are an important first step, the majority of this work relies on minimum inhibitory concentration (MIC) assays to assess the susceptibility patterns of reference strains to TAPI or its components (2, 4, 5, 14, 16, 20, 24–34). However, reference strains may not reflect the diversity of antimicrobial resistance or virulence mechanisms carried by pathogens currently circulating in the clinic today. Furthermore, *in vitro* conditions do not always recapitulate interactions that occur during infection. Specifically, for *S. aureus*, in addition to genetic resistance, these bacteria also exhibit phenotypic drug recalcitrance (35–41). Phenotypic recalcitrance among *S. aureus* strains typically involves the incorporation of various host factors present in the blood or during wound healing, such as fibrinogen and collagen, into biofilm structures (1, 28, 42). These biofilms provide recalcitrance to antibiotic concentrations at which their planktonic form is susceptible (4, 35, 36, 40, 41, 43–49). Moreover, these host proteins are available during breast surgery and may affect prophylactic antibiotic efficacy (24–27, 36, 43). Additionally, while TAPI prophylaxis has been associated with a reduction in the rate of capsular contracture, a latent complication thought to occur when low levels of bacteria contaminate host-formed capsules surrounding the prostheses, there is a lack of randomized, well controlled, clinical studies assessing the efficacy of TAPI at preventing overt BIAI (1, 2, 5, 9, 10, 15, 16, 50). Thus, it remains unclear how MIC assays using reference strains can be translated into effective irrigant strategies that prevent BIAIs.

In this study, we used *in vitro* and *in vivo* assays to assess the efficacy of TAPI against clinically relevant isolates of *S. aureus* to provide insights into how the pathogen resists antibiotics to become one of the most common causes of BIAI (1, 5). We characterized two *S. aureus* BIAI isolates (117 and 158), as well as JE2, a well studied reference strain. Using *in vitro* MIC assays, we found that all three *S. aureus* isolates displayed resistance to bacitracin but were susceptible to cefazolin and TAPI. Notably, JE2 also displayed resistance to gentamicin, while both BIAI isolates were susceptible to the antibiotic. Surprisingly, while biofilm formed by JE2 was not affected by exposure to TAPI *in vitro*, the irrigant enhanced the biomass of the BIAI strains compared to untreated controls. Furthermore, using a mouse model, we demonstrated that the *S. aureus* BIAI strain 117 displayed increased recalcitrance to TAPI compared to JE2. Specifically, JE2 was significantly reduced by TAPI prophylaxis compared to saline-treated controls at 1 and 7 days postinfection (dpi), while the irrigant had no effect on the ability of the BIAI strain

**TABLE 1** Strain information[a]

| Strain name | Isolation site | Genus | Species | Citations |
|---|---|---|---|---|
| JE2 | Skin infection | *Staphylococcus* | *aureus* | Bae et al. (51) |
| PAO1 | Pneumonia (cystic fibrosis) | *Pseudomonas* | *aeruginosa* | Vuong et al. (82) |
| 117 | BIAI | *Staphylococcus* | *aureus* | This study |
| 158 | BIAI | *Staphylococcus* | *aureus* | This study |
| 157 | BIAI | *Pseudomonas* | *aeruginosa* | This study |
| 160 | BIAI | *Pseudomonas* | *aeruginosa* | This study |

[a]BIAI, breast implant-associated infection.

117 to persist within tissues and on implants at the same time points. Notably, this phenotypic recalcitrance *in vivo* was unique to *S. aureus* clinical isolates, as when reference (PAO1) and BIAI isolates (157 and 160) of *P. aeruginosa* were assessed *in vitro*, and in the mouse model, TAPI was effective at killing the strains and preventing chronic infection. Thus, these data emphasize the disparity between clinical and reference isolates when determining the efficacy of prevention strategies. Furthermore, this study highlights how *S. aureus* BIAI isolates display unique mechanisms to resist antibiotics during BIAI.

## RESULTS

***S. aureus* antibiotic susceptibility patterns.** Two *S. aureus* strains isolated from BIAIs (117 and 158) and a reference strain (JE2) (Table 1) were assessed for their susceptibility to gentamicin, cefazolin, bacitracin, and TAPI, via MIC and minimum bactericidal concentration (MBC) assays (Fig. 1; Table S1). Both BIAI strains 117 and 158 display susceptibility to gentamicin, while JE2 exhibits resistance as expected (51) (Fig. 1A). All *S. aureus* strains display the same MBC of 1.0 $\mu$g/mL (Table S1). All *S. aureus* strains are susceptible to cefazolin and display MBCs of 6.0, 0.5, and 1.0 $\mu$g/mL for JE2, 117, and 158, respectively (Fig. 1B; Table S1). Additionally, all *S. aureus* strains display resistance to bacitracin and exhibit MBCs of >32 $\mu$g/mL (Fig. 1C; Table S1). We also assessed the susceptibility of all three *S. aureus* strains to the TAPI at the concentrations used in patients. TAPI was effective against all *S. aureus* strains, as there was no bacterial growth detected (Fig. 1D; Table S1). Lastly, the genomes of the BIAI strains 117 and 158 were sequenced, the sequence types (STs) were determined, and the potential antibiotic resistance genes were identified. While JE2 is an ST 8, the BIAI isolates 117 and 158 were STs 39 and 45, respectively (Table S2). Furthermore, while JE2 encodes the *mecA* gene, confirming that it is a methicillin-resistant *S. aureus* strain, neither BIAI isolate carried *mecA*, suggesting that both are methicillin susceptible (Table S2). Notably, the BIAI strains encode only a few genes with known roles in resistance to tetracycline (*tet-38* and *mepA/R*), supporting the MIC data above and suggesting they harbor limited acquired antibiotic resistance mechanisms.

***S. aureus* biofilms affect antibiotic efficacy.** All three *S. aureus* strains were assessed for biofilm formation and were able to form biofilm under standard *in vitro* conditions (Fig. 2). Additionally, biofilms were exposed to increasing concentrations of gentamicin and cefazolin, the antibiotics to which the BIAI isolates were susceptible during planktonic growth, including the MIC, twice the MIC, and four times the MIC. There was no significant difference in biomass among any of the strains when increasing concentrations of antibiotics were added to preformed biofilms (Fig. 2). However, TAPI significantly reduced the biomass of JE2 and the BIAI strain 158 (Fig. 2A and C). In contrast, the biomass of the BIAI strain 117 was not affected by TAPI (Fig. 2B). Additionally, biofilms were formed in the presence of human plasma to more closely mimic infection-like conditions, as previously described (29). Human plasma significantly enhanced biofilms formed by all *S. aureus* strains compared to those grown in media alone (Fig. 2 and 3). Additionally, when these biofilms were exposed to increasing concentrations of gentamicin and cefazolin there was no significant difference in biomass formed by any of the strains compared to unexposed controls, similarly to biofilms formed in media alone (Fig. 3A to C). However, in contrast to biofilms formed in media alone, TAPI had no effect on the biomass formed by JE2 under

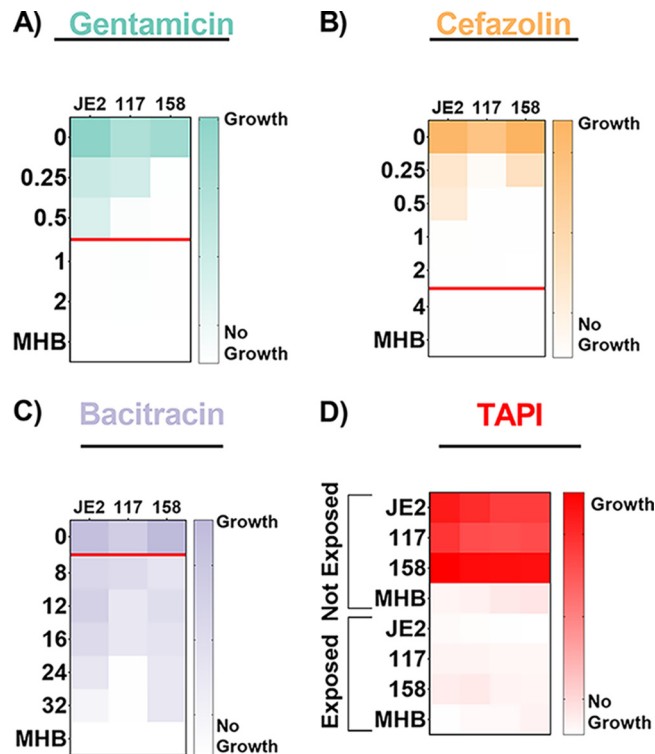

**FIG 1** Antibiotic susceptibility patterns of *S. aureus* strains. (A to C) Strains JE2, 117, and 158 were exposed to increasing concentrations of gentamicin (A), cefazolin (B), or bacitracin (C). (A) JE2 displays resistance to gentamicin, with a minimum inhibitory concentration (MIC) of 1.0 μg/mL, while 117 and 158 are susceptible, as they exhibit MICs of 0.5 and 0.25 μg/mL, respectively. (B) JE2, 117, and 158 are all susceptible to cefazolin, with MICs of 1.0, 0.5, and 0.5 μg/mL, respectively. (C) JE2, 117, and 158 are all resistant to bacitracin, as they exhibit MICs of >32, 24, and >32 μg/mL, respectively. The red line represents the MIC breakpoint for each antibiotic for *S. aureus*. An MIC value at or above the MIC breakpoint classifies the pathogen as resistant to the antibiotic. (D) Susceptibility of *S. aureus* strains to triple antibiotic pocket irrigant (TAPI) was determined based on an increase in optical density, indicating growth of strains, when exposed to TAPI compared to those not exposed to TAPI. JE2, 117, and 158 display susceptibility to TAPI. MICs were determined using an optical density at 600 nm ($OD_{600}$) threshold lower than 0.1, and heat maps display a representative of one of the three replicates. MHB, Mueller-Hinton broth.

these conditions (Fig. 3A). Notably, TAPI significantly increased the biomass formed by both of the BIAI isolates compared to control, unexposed biofilms under these conditions (Fig. 3B and C). Furthermore, to gain insights into potential virulence factors that might contribute to the increased biofilm formation observed with the BIAI strains, the virulence factor database (VFDB) was used in combination with our sequenced genomes (Fig. S1) (52). While this analysis indicated that there were not any differences in the carriage of genes with known roles in biofilm, such as polysaccharide intracellular adhesin, proteases, or clumping factor (Clf) A and B, it did show that the BIAI isolates encoded six genes with other roles in virulence that were absent in the JE2 strain, including the capsule genes *cap8H*, *cap8I*, *cap8J*, and *cap8K*; superantigen staphylococcal enterotoxin C (SEC); and staphylococcal enterotoxin-like type L (*selL*) (42, 53). It remains unclear whether any of these genes might contribute to the increased biofilm formation following exposure to TAPI in the presence of human plasma observed with the BIAI isolates or whether other unidentified factors are responsible.

**S. aureus displays recalcitrance to TAPI in a mouse BIAI model.** One representative *S. aureus* BIAI isolate (117) and the reference strain (JE2) were selected, and the efficacy of TAPI against these strains was assessed in a mouse model of BIAI (Fig. 4). For saline-treated control mice, high JE2 CFU were recovered from implants ($3.8 \times 10^5$ CFU), and the corresponding tissue samples ($3.4 \times 10^9$ CFU) were harvested at 1 dpi (Fig. 4A and B). This infection persisted at 7 dpi, as similar CFU were recovered from implants ($5.3 \times 10^5$ CFU)

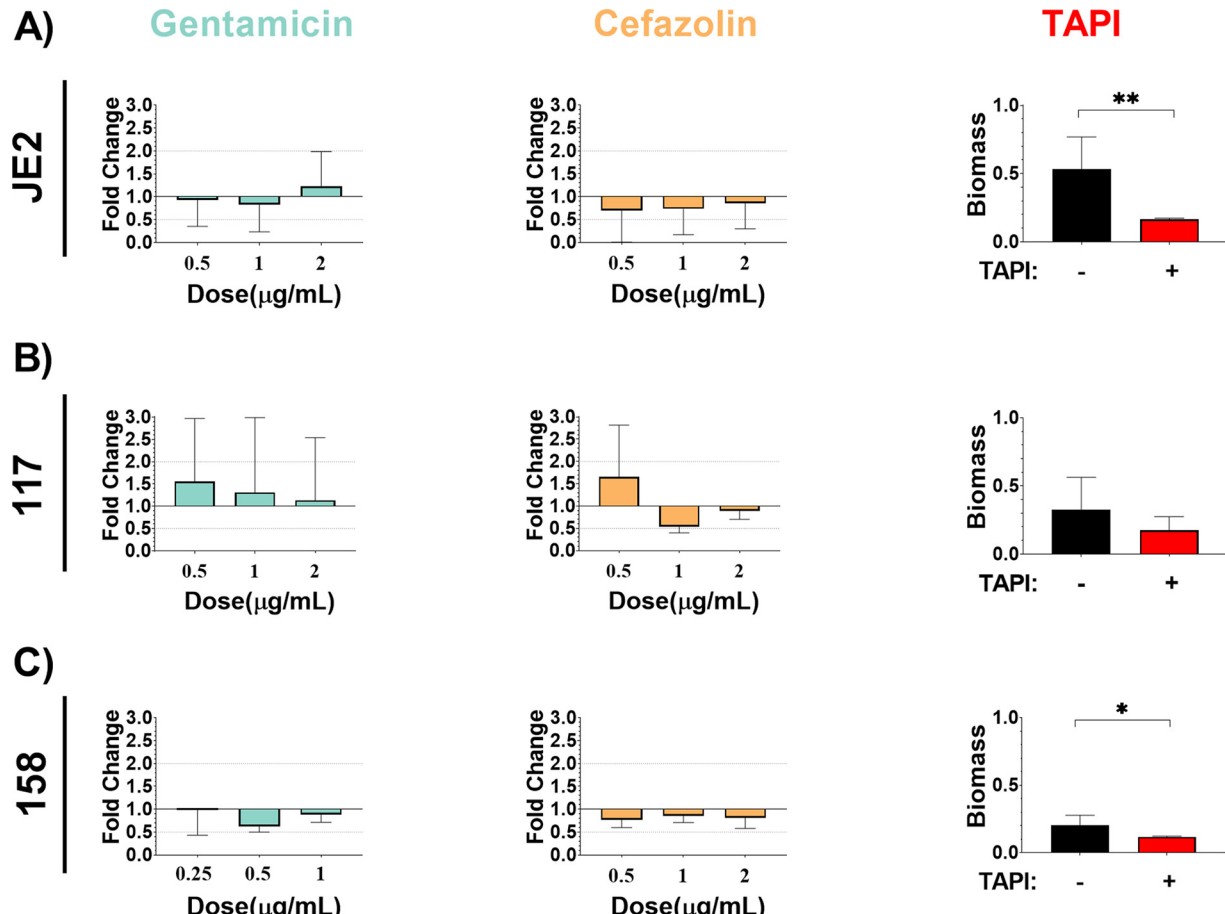

**FIG 2** *S. aureus* biofilms formed under *in vitro* conditions display recalcitrance to TAPI antibiotics. JE2 (A), 117 (B), and 158 (C) were allowed to form biofilm and were then exposed to increasing concentration of antibiotics. (A) JE2 biofilm (mean, 0.4406; standard deviation [SD], 0.2192; standard error of the mean [SE], 0.05315) was not significantly affected following exposure to increasing concentrations of gentamicin (0.5 μg/mL: *P* value, 0.9404; mean, 0.4083; SD, 0.2528; SE, 0.1460; and 1 μg/mL: *P* value, 0.5360; mean, 0.3653; SD, 0.2630; SE, 0.1518; and 2 μg/mL: *P* value, 0.5351; mean, 0.5410; SD, 0.3338; SE, 0.1927) or cefazolin (0.5 μg/mL: *P* value, 0.2314; mean, 0.3073; SD, 0.3052; SE, 0.1526; and 1 μg/mL: *P* value, 0.2314; mean, 0.3220; SD, 0.2488; SE, 0.1244; and 2 μg/mL: *P* value, 0.6869; mean, 0.3770; SD, 0.2464; SE, 0.1232). TAPI, however, significantly reduced the amount of biomass formed by JE2 (*P* value, 0.0018; mean, 0.1637; SD, 0.0140; SE, 0.003932). (B) 117 biofilm (mean, 0.2885; SD, 0.2274; SE, 0.06857) was not significantly affected following exposure to increasing concentrations of gentamicin (0.5 μg/mL: *P* value, 0.6608; mean, 0.4570; SD, 0.4167; SE, 0.2083; and 1 μg/mL: *P* value, 0.9714; mean, 0.3860; SD, 0.4954; SE, 0.2477; and 2 μg/mL: *P* value, 0.8491; mean, 0.3348; SD, 0.4128; SE, 0.2064), cefazolin (0.5 μg/mL: *P* value, 0.5549; mean, 0.4863; SD, 0.3433; SE, 0.1982; and 1 μg/mL: *P* value, 0.3429; mean, 0.1593; SD, 0.04207; SE, 0.02104; and 2 μg/mL: *P* value, 0.7325; mean, 0.2625; SD, 0.05548; SE, 0.02265) or to TAPI (*P* value, 0.3773; mean, 0.1746; SD, 0.1001; SE, 0.04477). (C) 158 biofilm (mean, 0.2013; SD, 0.07619; SE, 0.02694) was not significantly affected following exposure to increasing concentrations of gentamicin (0.25 μg/mL: *P* value, 0.6303; mean, 0.2217; SD, 0.1243; SE, 0.07178; and 0.5 μg/mL: *P* value, 0.4970; mean, 0.1410; SD, 0.02862; SE, 0.01652; and 1 μg/mL: *P* value, 0.8518; mean, 0.1987; SD, 0.03819; SE, 0.01559) or cefazolin (0.5 μg/mL: *P* value, 0.7758; mean, 0.1733; SD, 0.03911; SE, 0.02258; and 1 μg/mL: *P* value, 0.9507; mean, 0.1922; SD, 0.03274; SE, 0.01337; and 2 μg/mL: *P* value, 0.9433; mean, 0.1814; SD, 0.05110; SE, 0.02285). However, TAPI significantly reduced the amount of biomass formed by 158 (*P* value, 0.0485; mean, 0.1170; SD, 0.004761; SD, 0.002380). The Mann-Whitney U test was used to determine statistical significance, where ** indicates *P* < 0.01, and *** indicates *P* < 0.0005. *P* > 0.05 indicates no significant difference. Bar graphs include the means and SDs.

and tissues (3.3 × 10⁸ CFU) (Fig. 4A and B). However, at 14 dpi, only one of the recovered implants and about half of the tissues had detectable JE2 CFU (Fig. 4A and B). Compared to control mice, TAPI significantly reduced the JE2 CFU in the tissue at both 1 and 7 dpi, with 6.7 × 10⁸ and 1.9 × 10⁷ CFU recovered, respectively (Fig. 4B). At 14 dpi, about half of the TAPI-treated mice cleared JE2 from the tissue, similar to control mice. However, for implants, TAPI significantly reduced only the JE2 CFU at 1 dpi (Fig. 4A). There were no differences detected in JE2 CFU recovered from implants of TAPI-treated compared to control mice at 7 or 14 dpi.

For the BIAI strain 117, high CFU were recovered from implants (2.8 × 10⁵ CFU), and the corresponding tissue samples (8.5 × 10⁸ CFU) were harvested at 1 dpi from control mice (Fig. 4C and D). Notably, this infection persisted at 7 and 14 dpi, as similar CFU

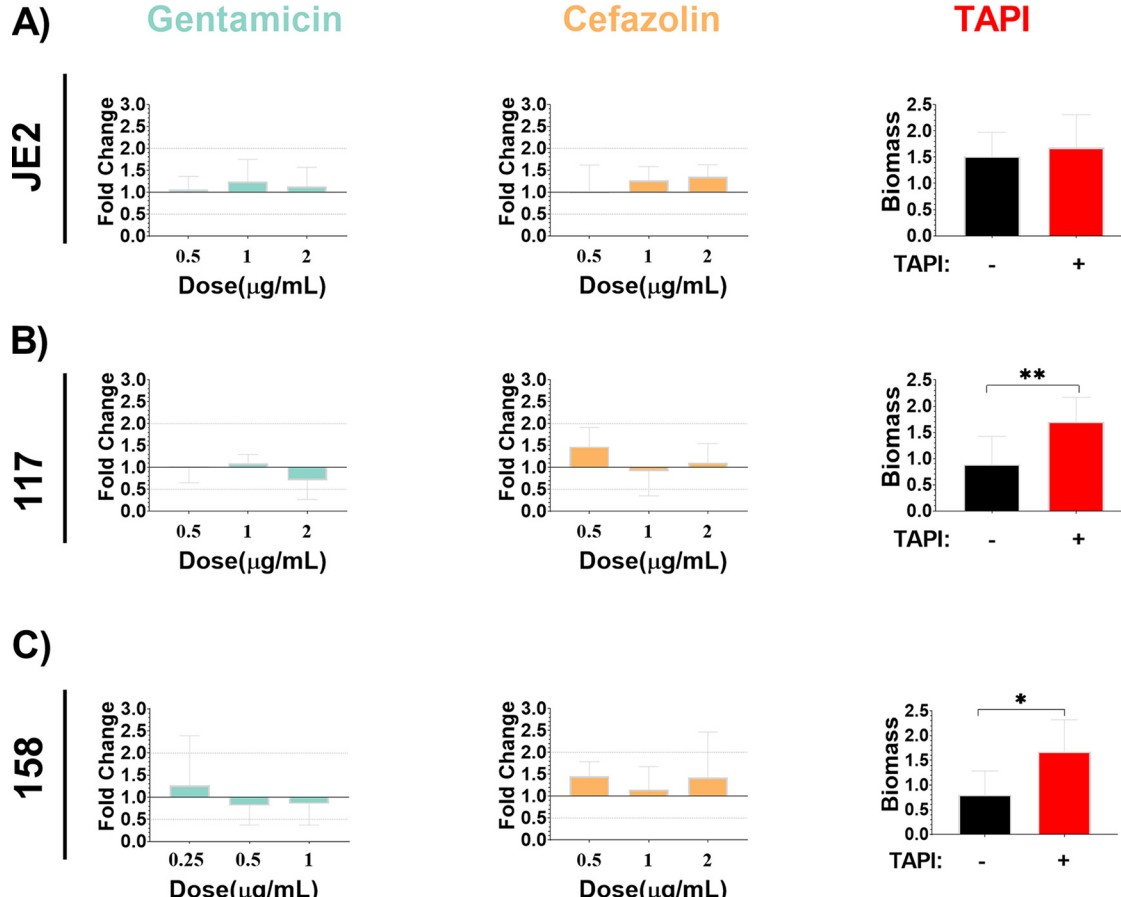

**FIG 3** *S. aureus* biofilms formed under *in vitro* conditions that mimic the host environment display recalcitrance to TAPI antibiotics. JE2 (A), 117 (B), and 158 (C) were allowed to form biofilm in the presence of human plasma and were then exposed to increasing concentrations of antibiotics. (A) JE2 biofilm (mean, 1.387; SD, 0.8289; SE, 0.2010) was not significantly affected following exposure to increasing concentrations of gentamicin (0.5 μg/mL: *P* value, 0.6895, mean, 1.473, SD, 0.4182, SE, 0.2414; 1 μg/mL: *P* value, 0.4158; mean, 1.718; SD, 0.7085; SE, 0.4091; and 2 μg/mL: *P* value, 0.6895; mean, 1.568; SD, 0.6093; SE, 0.3518), cefazolin (0.5 μg/mL: *P* value, 0.6977; mean, 1.404; SD, 0.8419; SE, 0.4209; 1 μg/mL: *P* value, 0.2750; mean, 1.763; SD, 0.4310; SE, 0.2155; and 2 μg/mL: *P* value, 0.1718; mean, 1.874; SD, 0.3805; SE, 0.1902) or to TAPI (*P* value, 0.4633; mean, 1.680; SD, 0.63060; SE, 0.2384). (B) 117 biofilm (mean, 0.9390; SD, 0.5298; SE, 0.1529) was not significantly affected following exposure to increasing concentrations of gentamicin (0.5 μg/mL: *P* value, 0.9527; mean, 0.9323; SD, 0.3203; SE, 0.1601; and 1 μg/mL: *P* value, 0.5989; mean, 1.027; SD, 0.1928; SE, 0.0963; and 2 μg/mL: *P* value, 0.5989; mean, 0.6678; SD, 0.4147; SE, 0.2073) or cefazolin (0.5 μg/mL: *P* value, 0.5549; mean, 1.381; SD, 0.4167; SE, 0.2406; and 1 μg/mL: *P* value, 0.6835; mean 0.8608; SD, 0.5292; SE, 0.2646; and 2 μg/mL: *P* value, 0.6820; mean, 1.041; SD, 0.4144; SE, 0.1692) but was significantly increased when exposed to TAPI (*P* value, 0.0268; mean, 1.702; SD, 0.4690; SE, 0.2098). (C) 158 biofilm (mean, 0.7978; SD, 0.4863; SE, 0.1621) was not significantly affected following exposure to increasing concentrations of gentamicin (0.25 μg/mL: *P* value, >0.9999; mean, 0.8670; SD, 0.7762; SE, 0.4481; and 0.5 μg/mL: *P* value, 0.7273; mean, 0.5587; SD, 0.3031; SE, 0.1750; and 1 μg/mL: *P* value, 0.5858; mean, 0.5858; SD, 0.3300; SE, 0.1347) or cefazolin (0.5 μg/mL: *P* value, 0.4818; mean, 0.9963; SD, 2.2337; SE, 0.1349; and 1 μg/mL: *P* value, 0.5827; mean, 0.7883; SD, 0.3636; SE, 0.1484; and 2 μg/mL: *P* value, 0.6064; mean, 0.9816; SD, 0.7121; SE, 0.3184) but was significantly increased when exposed to TAPI (*P* value, 0.0336; mean, 1.676; SD, 0.6467; SE, 0.3235). The Mann-Whitney U test was used to determine statistical significance, where ** indicates *P* < 0.01, and *** indicates *P* < 0.0005. *P* > 0.05 indicates no significant difference. Bar graphs include the mean and standard deviation.

were recovered from implants ($5.0 \times 10^4$ CFU and $6.3 \times 10^4$ CFU, respectively) and tissues ($2.3 \times 10^7$ CFU and $7.6 \times 10^6$ CFU, respectively) (Fig. 4C and D). Importantly, TAPI was ineffective at reducing the CFU within either the implants or tissues of the mice infected with the BIAI strain 117 at any time point tested (Fig. 4C and D). Thus, while TAPI treatment is effective against JE2, which is cleared by the mice over time, the BIAI 117 strain resists the irrigant, resulting in chronic infection.

**_P. aeruginosa_ antibiotic susceptibility patterns.** To determine whether TAPI recalcitrance was unique to *S. aureus* BIAI isolates or whether other species causing these infections also display a similar phenotype, TAPI efficacy was assessed against *P. aeruginosa* strains (Table 1). *P. aeruginosa* BIAI isolates (157 and 160) and a reference strain (PAO1) were assessed for their susceptibility to gentamicin, cefazolin, bacitracin, and

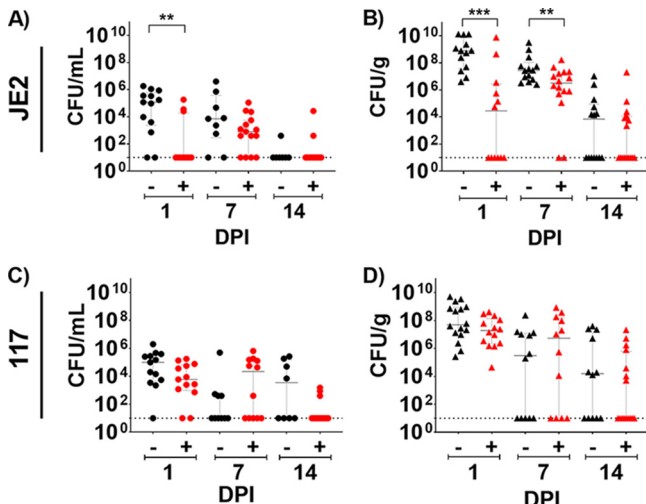

**FIG 4** Mouse model of *S. aureus* breast implant-associated infection (BIAI). (A) Implants recovered from saline-treated mice displayed high colony forming units (CFU) of JE2 at 1 day postinfection (dpi) (mean, $3.8 \times 10^5$; SD, $5.65 \times 10^5$; SE, $1.57 \times 10^5$), which persisted at 7 dpi (mean, $5.32 \times 10^5$; SD, $1.32 \times 10^6$; SE, $4.41 \times 10^5$). However, mice were able to control JE2 infection by 14 dpi (mean, $6.57 \times 10^1$; SD, $1.47 \times 10^2$; SE, $5.57 \times 10^1$). Triple antibiotic pocket irrigant (TAPI) significantly reduced JE2 CFU on implants compared to the saline-treated control mice at 1 dpi (*P* value, 0.0030; mean, $2.05 \times 10^4$; SD, $5.29 \times 10^4$; SE, $1.53 \times 10^4$). However, by 7 dpi (*P* value, 0.2049; mean, $1.17 \times 10^4$; SD, $2.91 \times 10^4$; SE, $7.51 \times 10^3$), there was no difference in CFU recovered from TAPI or saline-treated mice. At 14 dpi (*P* value, >0.9999; mean, $1.82 \times 10^3$; SD, $6.89 \times 10^3$; SE, $1.78 \times 10^3$), most mice cleared JE2 regardless of treatment. (B) Tissue from saline-treated mice displayed high CFU of JE2 at 1 and 7 dpi (mean, $3.43 \times 10^9$; SD, $5.44 \times 10^9$; SE, $1.51 \times 10^9$; and mean, $3.32 \times 10^8$; SD, $8.63 \times 10^8$; SE, $2.31 \times 10^8$, respectively). However, half the mice were able to control the infection by 14 dpi (mean, $9.50 \times 10^5$; SD, $2.83 \times 10^6$; SE, $7.56 \times 10^5$). TAPI significantly reduced CFU in the tissue compared to the saline-treated mice at 1 and 7 dpi (*P* value, 0.0002; mean, $6.72 \times 10^8$; SD, $2.19 \times 10^9$; SE, $6.32 \times 10^8$; and *P* value, 0.0089; mean, $1.86 \times 10^7$; SD, $4.10 \times 10^7$; SE, $1.03 \times 10^7$, respectively). Again, about half the mice cleared the infection with JE2 regardless of treatment by 14 dpi (*P* value, 0.4409; mean, $1.27 \times 10^6$; SD, $5.05 \times 10^8$; SE, $1.26 \times 10^6$). (C) Implants recovered from saline-treated mice displayed high CFU of 117 at 1 dpi (mean, $2.84 \times 10^5$; SD, $5.40 \times 10^5$; SE, $1.50 \times 10^5$), which persisted at both 7 and 14 dpi (mean, $4.97 \times 10^4$; SD, $1.57 \times 10^5$; SE, $4.96 \times 10^4$; and mean, $6.31 \times 10^4$, SD, $1.03 \times 10^5$; SE, $3.64 \times 10^4$, respectively). There was no significant difference in CFU recovered from implants of TAPI-treated mice at 1 dpi (*P* value, 0.0640; mean, $3.87 \times 10^4$; SD, $5.81 \times 10^4$; SE, $1.61 \times 10^4$), 7 dpi (*P* value, 0.2481; mean, $1.01 \times 10^5$; SD, $1.85 \times 10^5$; SE, $5.34 \times 10^4$), and 14 dpi (*P* value, 0.0557; mean, $1.95 \times 10^2$; SD, $4.37 \times 10^2$; SE, $1.13 \times 10^2$) compared to saline-treated mice. (D) Tissue from saline-treated mice displayed high CFU of 117 at 1 dpi (mean, $8.35 \times 10^7$; SD, $1.28 \times 10^8$; SE, $3.43 \times 10^7$), which persisted at both 7 and 14 dpi (mean, $1.25 \times 10^8$; SD, $2.49 \times 10^8$; SE, $7.18 \times 10^7$; and mean, $1.80 \times 10^6$; SD, $5.31 \times 10^6$; SE, $1.33 \times 10^6$, respectively). There was no difference in CFU recovered from tissues of TAPI-treated mice at 1 dpi (*P* value, 0.1417; mean, $8.35 \times 10^7$; SD, $1.28 \times 10^8$; SE, $3.43 \times 10^7$), 7 dpi (*P* value, 0.3638; mean, $1.25 \times 10^8$; SD, $2.49 \times 10^8$; SE, $7.18 \times 10^7$), and 14 dpi (*P* value, 0.2980; mean, $1.80 \times 10^6$; SD, $5.31 \times 10^7$; SE, $1.33 \times 10^7$) compared to saline-treated mice. Red represents TAPI-treated mice, while black denotes saline-treated mice. Each circle represents the CFU recovered from an implant of each mouse. Each triangle denotes the CFU retrieved from tissue near the implant of each mouse. The Mann-Whitney U test was used to determine statistical significance, where ** indicates $P < 0.01$, and *** indicates $P < 0.0005$. $P > 0.05$ indicates no significant differences. Bar graphs represent the median and the interquartile range.

TAPI via MIC and MBC assays (Fig. 5; Table S1). All of the *P. aeruginosa* strains display susceptibility to gentamicin and exhibit MBCs of 6, 4, and 6 $\mu$g/mL for PAO1, 157, and 160, respectively (Fig. 5; Table S1). However, all *P. aeruginosa* strains are resistant to cefazolin and bacitracin and display MBCs of >32 $\mu$g/mL for cefazolin and >1,400 $\mu$g/mL for bacitracin for all three isolates (Fig. 5B and C; Table S1). For all *P. aeruginosa* strains exposed to TAPI, no bacterial growth was detected, indicating that the bacteria are susceptible to this combination of antibiotics *in vitro* (Fig. 5D; Table S1). Lastly, the genomes of the BIAI strains 157 and 160 were sequenced, the STs were determined, and the potential antibiotic resistance genes were identified. While PAO1 is a ST 549, both BIAI strains were ST 633 (Table S2). Furthermore, both BIAI strains encode genes with known roles in resistance to cephalosporins (*oxa-486*), supporting the MIC data for cefazolin above.

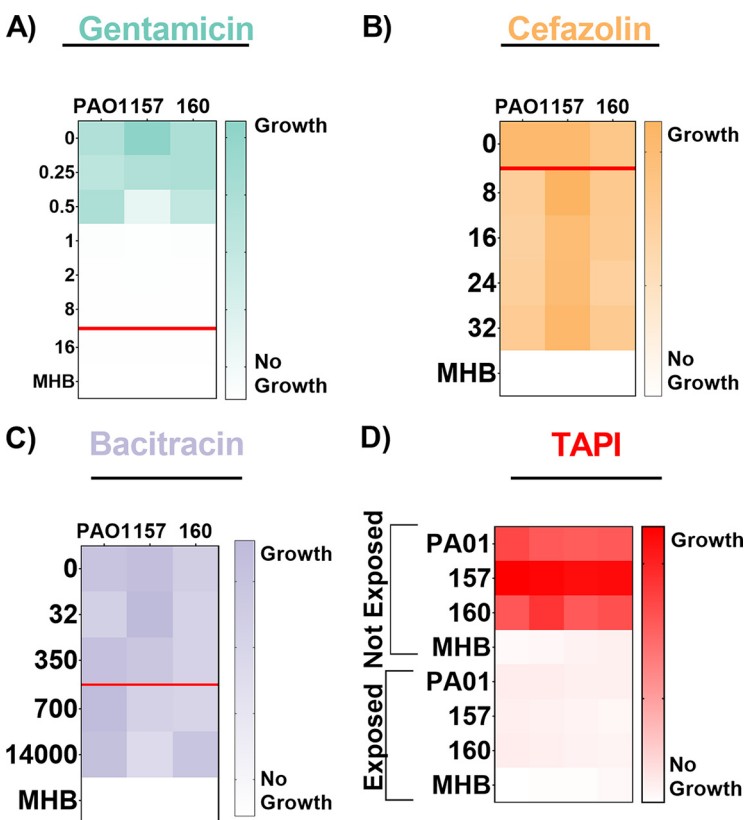

**FIG 5** Antibiotic susceptibility patterns of *P. aeruginosa* strains. PAO1, 157, and 160 strains were exposed to increasing concentrations of gentamicin (A), cefazolin (B), or (C) bacitracin. (A) PAO1, 157, and 160 all displayed minimum inhibitory concentrations (MICs) of 1 $\mu$g/mL, indicating that they are susceptible to gentamicin. (B) PAO1, 157, and 160 all displayed MICs of $>$32 $\mu$g/mL, indicating that they are all resistant to cefazolin. (C) PAO1, 157, and 160 all displayed MICs of $>$1,400 $\mu$g/mL, indicating that they are all resistant to bacitracin. The red line represents the MIC breakpoint for each antibiotic for *P. aeruginosa*. A MIC value at or above the MIC breakpoint classifies the pathogen as resistant to the antibiotic. (D) Susceptibility of *P. aeruginosa* strains to triple antibiotic pocket irrigant (TAPI) was determined based on an increase in optical density, indicating growth of strains, when exposed to TAPI compared to those not exposed. PAO1, 157, and 160 are susceptible to TAPI. The MICs were determined using an $OD_{600}$ threshold lower than 0.1, and the heat maps display a representative of one of the three replicates.

**TAPI is effective against communities formed by *P. aeruginosa*.** The three *P. aeruginosa* strains were assessed for biofilm and aggregate formation using previously published conditions (4, 35, 43). The *P. aeruginosa* strains were able to form biofilms and aggregates (Fig. 6). Additionally, biofilms and aggregates were exposed to increasing concentrations of gentamicin, the antibiotic to which the BIAI isolates were susceptible during planktonic growth, including the MIC, twice the MIC, and four times the MIC. Biofilm formed by the historical strain PAO1 was significantly reduced in the presence of 2 and 4 $\mu$g/mL of gentamicin (Fig. 6A). Additionally, aggregates were also significantly reduced in the presence of 2 $\mu$g/mL of gentamicin (Fig. 6B). Biofilm formed by the BIAI strain 157 was also significantly reduced at 4 $\mu$g/mL of gentamicin (Fig. 6C). However, the aggregates were unaffected by any concentration of gentamicin tested (Fig. 6D). Lastly, while the biofilm formed by the BIAI strain 160 was recalcitrant to all the concentrations of gentamicin tested (Fig. 6E), the aggregates were significantly reduced at 4 $\mu$g/mL (Fig. 6F). Notably, TAPI significantly reduced biofilm formation of PAO1, 157, and 160, indicating that the antibiotics combined at the concentration used in TAPI were effective at killing *P. aeruginosa* (Fig. 6A, C, and E).

**TAPI is effective against *P. aeruginosa* in a mouse model of BIAI.** To investigate the efficacy of TAPI against *P. aeruginosa*, the reference strain (PAO1) and one BIAI isolate (157) were assessed in the mouse BIAI model. For control mice, high PAO1 CFU were recovered from implants and the corresponding tissue samples at 1 and 7 dpi (Fig. 7A and

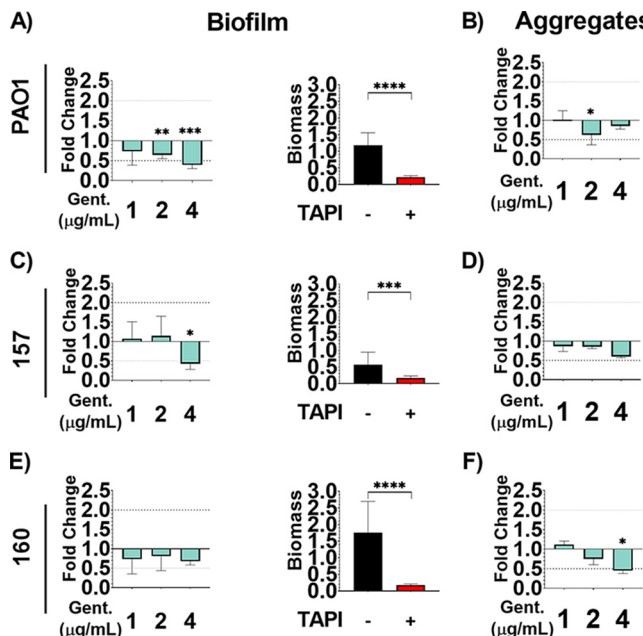

**FIG 6** Communities formed by *P. aeruginosa* are susceptible to the triple antibiotic irrigant (TAPI) antibiotics. *P. aeruginosa* strains were allowed to form biofilms and aggregates and were then exposed to increasing concentrations of gentamicin, the antibiotic at which the planktonic bacteria were susceptible. (A) 2 and 4 $\mu$g/mL (twice and four times the minimum inhibitory concentration [MIC], respectively) of gentamicin (*P* value, 0.0082; mean, 0.7618; SD, 0.1118; SE, 0.04563; and *P* value, 0.0002; mean, 0.4638; SD, 0.1091; SE, 0.04452, respectively) significantly reduced the biomass of PAO1 biofilms (mean, 1.186; SD, 0.3695; SE, 0.09874), while the 1 $\mu$g/mL (MIC) concentration (*P* value, 0.0672; mean, 0.8623; SD, 0.33811, SE, 0.1348) did not cause a significant change. TAPI also significantly reduced PAO1 biofilm (*P* value, <0.0001; mean, 0.2180; SD, 0.04770; SE, 0.01803). (B) 2 $\mu$g/mL (twice the MIC) of gentamicin (*P* value, 0.0500; mean, 1.745; SD, 0.7246; SE, 0.4184) significantly reduced PAO1 aggregate biomass (mean, 2.804; SD, 0.7165; SE, 0.4137), while the 1 $\mu$g/mL (MIC) concentration (*P* value, 0.5000; mean, 2.848; SD, 0.6535; SE, 0.3773) and the 4 $\mu$g/mL (twice the MIC) concentration (*P* value, 0.2000; mean, 2.393; SD, 0.2317; SE, 0.1338) did not. (C) 4 $\mu$g/mL (four times the MIC) of gentamicin (*P* value, 0.0117; mean, 0.2412; SD, 0.08226; SE, 0.03358) significantly reduced the biomass of the biofilm form by the BIAI strain 157 (mean, 0.5656; SD, 0.3793; SE, 0.1014), while 1 $\mu$g/mL (MIC) and 2 $\mu$g/mL (twice the MIC) did not cause a significant change in biomass (*P* value, 0.4020; mean, 0.6021; SD, 0.2303; SE, 0.08144; and *P* value, 0.4334; mean, 0.6453; SD, 0.2885; SE, 0.1178, respectively). TAPI also significantly reduced the biofilm biomass (*P* value, 0.0006; mean, 0.1757; SD, 0.05315; SE, 0.02170). (D) No concentration of gentamicin (1 $\mu$g/mL [MIC]: *P* value, 0.5000; mean, 1.696; SD, 0.2623; SE, 0.1514; and 2 $\mu$g/mL [twice the MIC]: *P* value, 0.2000; mean, 1.677; SD, 0.09586; SE, 0.05535; and 4 $\mu$g/mL [four times the MIC]: *P* value, 0.1000; mean, 1.181; SD, 0.06364; SE, 0.04500) tested affected the biomass of aggregates formed by the BIAI strain 157 (mean, 1.965; SD, 0.4182; SE, 0.2415). (E) No concentration of gentamicin (1 $\mu$g/mL: *P* value, 0.3650; mean, 1.286; SD, 0.6711; SE, 0.2373; and 2 $\mu$g/mL: *P* value, 0.6590; mean, 1.414; SD, 0.6464; SE, 0.2639; and 4 $\mu$g/mL: *P* value, 0.4940; mean, 1.95; SD, 0.1758; SE, 0.07176) tested affected the biofilm formed by the BIAI strain 160 (mean, 1.753; SD, 0.9438; SE, 0.2522). However, TAPI significantly reduced the biofilm biomass (*P* value < 0.0001; mean, 0.1828; SD, 0.03024; SE, 0.1234). (F) 4 $\mu$g/mL (four times the MIC) of gentamicin (4 $\mu$g/mL: *P* value, 0.0500; mean, 1.912; SD, 0.3149; SE, 0.1818) significantly reduced the biomass of the aggregates formed by the BIAI strain 160 (mean, 4.241; SD, 1.225; SE, 1.7073), while 1 $\mu$g/mL (MIC) (*P* value, 0.4000; mean, 4.735; SD, 0.3771; SE, 0.217) and 2 $\mu$g/mL (twice the MIC) (*P* value, 0.2000; mean, 3.184; SD, 0.6403; SE, 0.3697) did not. The Mann-Whitney U test was used to determine statistical significance, where ** indicates *P* < 0.01, and *** indicates *P* < 0.0005. *P* > 0.05 indicates no significant differences. The bar graphs include the means and SD.

B). However, at 14 dpi, control mice largely eliminated PAO1 from implants and tissues. Notably, TAPI significantly reduced the PAO1 CFU recovered from implants and tissues at 1 dpi (Fig. 7A and B). At 7 dpi, TAPI also significantly reduced PAO1 CFU recovered from implants compared to control mice (Fig. 7A). In contrast, however, PAO1 CFU recovered from tissues of TAPI-treated mice were similar to those from control mice at the same time point (Fig. 7B). By 14 dpi, similar CFU were recovered from both implants and tissues of TAPI-treated mice compared to control mice (Fig. 7A and B). For the *P. aeruginosa* BIAI strain 157, $1.0 \times 10^3$ and $8.1 \times 10^5$ CFU were recovered from implants and the corresponding tissue samples of control mice, respectively, at 1 dpi (Fig. 7C and D). At 7 dpi,

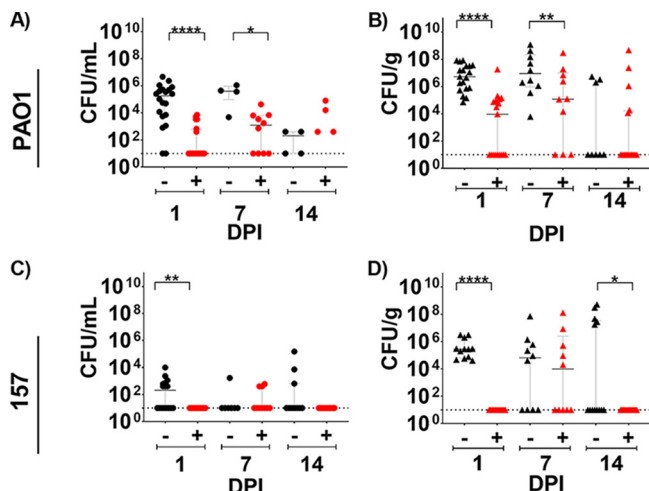

**FIG 7** *P. aeruginosa* mouse breast implant-associated infection (BIAI) model. (A) Implants recovered from saline-treated, control mice displayed high colony forming units (CFU) of PAO1 at 1 day postinfection (dpi) (mean $5.69 \times 10^5$; SD $1.11 \times 10^6$; SE $2.47 \times 10^5$), which persisted at 7 dpi (mean, $4.78 \times 10^5$; SD, $4.56 \times 10^5$; SE, $2.28 \times 10^5$). However, mice were able to eliminate PAO1 by 14 dpi (mean, $2.05 \times 10^2$; SD, $2.25 \times 10^2$; SE, $1.13 \times 10^2$). TAPI significantly reduced CFU on implants compared to control mice at 1 and 7 dpi (*P* value, <0.0001; mean, $1.03 \times 10^3$; SD, $2.09 \times 10^3$; SE, $4.67 \times 10^2$; and *P* value, 0.0120; mean, $6.16 \times 10^3$; SD, $1.30 \times 10^4$; SE, $4.11 \times 10^3$, respectively). At 14 dpi, most mice eliminated PAO1 regardless of treatment (*P* value, 0.1504; mean, $7.45 \times 10^3$; SD, $2.22 \times 10^4$; SE, $6.17 \times 10^3$). (B) Tissue from control mice displayed high CFU of PAO1 at 1 dpi (mean, $2.05 \times 10^7$; SD, $2.79 \times 10^7$; SE, $6.39 \times 10^5$), which persisted at 7 dpi (mean, $1.79 \times 10^8$; SD, $3.68 \times 10^8$; SE, $1.16 \times 10^8$). At 14 dpi, most mice eliminated PAO1 from the tissue (mean, $1.32 \times 10^6$; SD, $2.03 \times 10^6$; SE, $7.18 \times 10^5$). TAPI significantly reduced PAO1 CFU in the tissue compared to control mice at 1 dpi (*P* value, <0.0001; mean, $1.00 \times 10^6$; SD, $4.17 \times 10^6$; SE, $9.56 \times 10^5$). However, at 7 dpi, similar CFU were recovered from the tissue of TAPI-treated compared to control mice (*P* value, 0.0748; mean, $3.19 \times 10^7$; SD, $9.12 \times 10^7$; SE, $2.88 \times 10^7$). At 14 dpi, most TAPI-treated mice eliminated PAO1 from the tissue, which was similar to control mice (*P* value, 0.7327; mean, $3.03 \times 10^7$; SD, $1.15 \times 10^8$; SE, $2.87 \times 10^7$). (C) CFU of BIAI strain 157 could be detected on implants recovered from control mice at 1 dpi (mean, $1.00 \times 10^3$; SD, $2.48 \times 10^3$; SE, $6.19 \times 10^2$). However, at 7 and 14 dpi, most control mice eliminated 157 from the implants (mean, $2.49 \times 10^2$; SD, $6.31 \times 10^2$; SE, $2.39 \times 10^2$; and mean, $1.45 \times 10^4$; SD, $4.56 \times 10^4$; SE, $1.38 \times 10^4$, respectively). TAPI significantly reduced the 157 CFU compared to control mice, as no bacteria were recovered from implants of TAPI-treated mice at 1 and 14 dpi (*P* value, 0.0057; mean, 10; SD, 0; SE, 0; and *P* value, 0.0159; mean, 10; SD, 0; SE, 0, respectively), while only a few TAPI-treated mice had detectable CFU on implants at 7 dpi (*P* value, 0.6691; mean, $1.47 \times 10^2$; SD, $2.27 \times 10^2$; SE, $7.19 \times 10^1$). (D) Tissue from control mice displayed high CFU of BIAI strain 157 at 1 and 7 dpi (mean, $8.12 \times 10^5$; SD, $1.19 \times 10^6$; SE, $3.42 \times 10^5$ and mean, $7.39 \times 10^6$; SD, $2.26 \times 10^7$; SE, $7.14 \times 10^6$, respectively). However, more than half of the control mice had no detectable 157 CFU at 14 dpi (mean, $6.82 \times 10^7$; SD, $1.54 \times 10^8$; SE, $4.12 \times 10^7$). TAPI significantly reduced CFU in the tissue compared to the control mice at 1 and 14 dpi (*P* value, <0.0001; mean, 1.00E1; SD, 0; SE, 0; and *P* value, 0.0159; mean, 1.00E1; SD, 0; SE, 0, respectively). While there was no difference in CFU between TAPI and control mice at 7 dpi, few mice had detectable CFU of 157 in the tissue (*P* value, 0.7863; mean, $1.39 \times 10^7$; SD, $4.09 \times 10^7$, SE, $1.29 \times 10^7$). Red represents TAPI-treated mice, while black denotes control mice. Each circle represents the CFU recovered from an implant of each mouse. Each triangle denotes the CFU retrieved from tissue near the implant of each mouse. The Mann-Whitney U test was used to determine statistical significance, where ** indicates *P* < 0.01, and *** indicates *P* < 0.0005. *P* > 0.05 indicates no significant difference. Graphs represent the median and the interquartile range.

only one implant had detectable CFU, while more of the tissue samples had detectable CFU at the same time point (Fig. 7C and D). At 14 dpi, only three implants and five of the corresponding tissue samples had detectable CFU (Fig. 7C and D). Notably, TAPI was effective at preventing infection with the BIAI strain 157, as no bacteria were recovered from implants or tissue from 1 or 14 dpi (Fig. 7C and D). However, at 7 dpi, low CFU of the BIAI strain 157 were recovered from only a few implants and tissue samples of TAPI-treated mice (Fig. 7C and D). Interestingly, there was no difference between CFU recovered from tissues of the control mice compared to the TAPI-treated mice infected with either PAO1 or the BIAI strain 157 at 7 dpi (Fig. 7B and D). Importantly, TAPI significantly reduced PAO1 and the BIAI strain 157 at early time points, with the BIAI strain 157 displaying increased susceptibility to TAPI, as the mice were successful at eliminating the infection by 14 dpi.

## DISCUSSION

Up to a third of all prostheses placed in immediate postmastectomy breast reconstruction annually become infected, despite sterile surgical techniques and the use of infection prevention strategies, such as surgical skin scrubs, pre- and postoperative antibiotic administration, and TAPIs (1, 2, 4–7). Furthermore, BIAIs are extremely difficult to treat, as they result in recalcitrant biofilm-associated infections that resist antibiotic therapies and require explantation of the infected prosthesis for complete resolution (1, 2, 5, 7). Thus, the implementation of evidence-based strategies that effectively prevent BIAIs have become a priority. This study focuses on assessing the efficacy of a commonly used prevention method, TAPI, against some of the most common causes of BIAIs, *S. aureus* and *P. aeruginosa*. We used recently isolated, clinically relevant strains to gain insights into the genomic and phenotypic antibiotic resistance mechanisms of currently circulating isolates causing BIAIs.

Antibiotic susceptibility testing revealed that the *S. aureus* BIAI isolates were susceptible to two of the antibiotics that make up TAPI (gentamicin and cefazolin) but resistant to the third (bacitracin). Additionally, while the *P. aeruginosa* BIAI strains were similarly resistant to bacitracin, these isolates also exhibited resistance to a second antibiotic in TAPI, cefazolin. Using whole-genome sequencing (WGS) and bioinformatics analyses, we found that the *S. aureus* BIAI isolates carried only a few acquired resistance genes, which mostly provide resistance to tetracycline, an antibiotic not used in TAPI (54). In contrast, the reference *S. aureus* strain JE2 encodes the *mecA* and *lmr* genes, which provide resistance to $\beta$-lactam and aminoglycoside antibiotics, respectively (55–57). Interestingly, all three *P. aeruginosa* strains encoded similar genes that provide resistance to cephalosporins, including *oxa*-50 for PAO1 and *oxa-486* for both BIAI strains (58–60). Thus, the genomic analyses support our MIC data. Importantly, while all the *S. aureus* and *P. aeruginosa* strains assessed exhibited resistance to at least one antibiotic, combining the drugs at the concentrations used in TAPI was effective at killing all the strains grown under planktonic conditions, which supports previous *in vitro* work (4, 14). Because the current antibiotic concentrations in TAPI are extremely high—160, 500, and 208 times the MIC of gentamicin, cefazolin, and bacitracin, respectively, for *S. aureus*; and 10, 250, and 2 times the MIC for gentamicin, cefazolin, and bacitracin, respectively, for *P. aeruginosa*—it is not surprising that TAPI is effective *in vitro*. However, administering an antibiotic (even at these exceedingly high concentrations) to treat infections caused by strains with known resistance to that antibiotic demonstrates a lack of antibiotic stewardship. Furthermore, while this study tests only a limited number of isolates, the fact that even the reference *S. aureus* and *P. aeruginosa* strains display resistance to bacitracin calls into question why this antibiotic is included in TAPI. Specifically, antimicrobial stewardship dictates that if an infecting pathogen is resistant to an antibiotic or if there is a high incidence of resistance among certain patient populations, that specific antibiotic should not be used to treat that infection as treatment failure is likely to occur (61–63).

By assessing TAPI efficacy *in vivo*, our results provide support for the importance of antibiotic stewardship guidelines. Specifically, by selecting representative *P. aeruginosa* and *S. aureus* BIAI isolates and comparing the infection phenotypes to reference strains, we demonstrated that a surprising number of these bacteria could persist despite TAPI treatment. While the *P. aeruginosa* BIAI strain 157 was the most susceptible to TAPI and resulted in the bacteria being eliminated from implants and tissues, TAPI significantly reduced the CFU of only the reference strains—JE2 and PAO1—in the samples at early time points. Furthermore, almost all mice infected with either reference strain maintained high CFU over a 7-day time course, regardless of treatment. However, by 14 dpi, about half the mice began to control the infection, regardless of treatment. These results correspond with a previous study using a mouse model of skin infection that showed a different historical *S. aureus* strain followed a similar trend of spontaneous elimination of the bacteria by 14 dpi (64). In contrast, however, the *S. aureus* BIAI strain 117 displayed complete recalcitrance to TAPI, with high CFU from harvested implants or tissues and no significant difference in CFU recovered from

samples of TAPI-treated compared to control mice over a 14-day time course. Together, these data suggest that TAPI is effective at reducing CFU only at early time points after surgery and that any bacteria that persist in the presence of TAPI can go on to cause chronic infection. Additionally, these data suggest that while TAPI may have some efficacy against *P. aeruginosa* strains causing BIAI, as BIAI strain 157 was fairly susceptible to TAPI in the mouse model, the irrigant may not be as efficacious at preventing BIAI with *S. aureus*, as the *S. aureus* isolates, and particularly the BIAI strain 117, were uniquely persistent over a 14-day time course. Most importantly, these data highlight the discrepancies between *in vitro* and *in vivo* results and suggest that they may not always accurately inform the efficacy of prevention strategies.

In seeking to understand why the *S. aureus* strains displayed increased recalcitrance *in vivo* compared to the *P. aeruginosa* isolates, despite exhibiting increased susceptibility to the antibiotics in TAPI, we assessed the ability of these strains to form community structures. The most well studied bacterial communities are biofilms, which promote recalcitrance to antibiotics as well as the host immune system (35, 36, 43, 44). All *S. aureus* and *P. aeruginosa* isolates were able to form biofilm under *in vitro* conditions. Importantly, these biofilms provided increased recalcitrance to antibiotic concentrations at which the planktonic bacteria were susceptible. Fortunately, TAPI was effective at significantly reducing the biomass of the biofilm formed by all three *P. aeruginosa* strains. In contrast, *S. aureus* biofilm and TAPI recalcitrance was affected by the conditions used to form biofilm. Specifically, using an *in vitro* biofilm assay that more closely mimics the host environment during infection, we demonstrated that when *S. aureus* formed biofilm in the presence of human plasma, the biomass increased, along with the pathogen's recalcitrance to TAPI. *S. aureus* is known to exploit host proteins released as part of the inflammatory and wound healing pathway, such as fibrinogen and collagen, and incorporate them into their biofilm structure to promote recalcitrance (36, 65–67). These host proteins, which accumulate within the breast during the surgical procedures required for implantation of breast prostheses and attach to the device surface, create a suitable environment for *S. aureus* biofilm formation during BIAI (24, 25, 36). Furthermore, it is well known that antibiotics themselves can act as signals to induce biofilm formation or promote drug tolerance (68–70). To understand the mechanisms contributing to increased biofilm formation and recalcitrance to TAPI, we used our whole-genome sequences and the VFDB of known virulence determinants to identify potential genes present in the BIAI isolates but absent in JE2 that may contribute to the observed phenotype (52). While capsule and enterotoxin-like genes were present in both BIAI strains and absent in JE2, it is unclear how these genes may enhance biofilm and/or recalcitrance to TAPI. Other genes with known roles in biofilm, adhesion to the host, and immune evasion, including polysaccharide intercellular adhesin, ClfA and ClfB, von Willebrand factor binding protein (vWbp), fibronectin binding protein A (*fnbA*), α-toxin (*hla*), hyaluronidase (*hysA*), staphylokinase (*sak*), chemotaxis inhibitory protein (*chp*), and staphylococcal complement inhibitor (*scn*), were present in all three strains. Thus, these data suggest that *S. aureus* isolates causing BIAI may encode unique mechanisms that respond to host proteins or antibiotic stimuli to promote recalcitrance during infection compared to reference strains, which are commonly used for these types of studies.

This study demonstrates that reference strains can exhibit fundamentally different phenotypes compared to clinically relevant, currently circulating BIAI isolates. Specifically, while the *S. aureus* BIAI strain 117 displayed similar MICs to the reference strain JE2, the BIAI isolate displayed increased recalcitrance to TAPI during biofilm formation and *in vivo*. The discovery that TAPI may increase the biofilm biomass of the *S. aureus* BIAI isolates has important implications for prophylactic antibiotic treatment in breast implant-based reconstructive surgeries. An increase in bacterial community formation, particularly in the face of antibiotic prophylaxis strategies intended to mitigate breast implant bacterial contamination, has significant clinical impact. The majority of plastic surgeons use TAPI or other antiseptics like povidone iodine to irrigate the surgical site when placing a breast implant

with the intention of limiting bacterial infection (71–73). Despite the use of antibiotic irrigation, parenteral antibiotics, surgical drains, and meticulous operative technique, the breast implant surface may remain chronically contaminated with bacteria *in vivo* in a clinically benign setting (30). However, in an effort to irradicate bacteria, plastic surgeons may be inadvertently selecting for more resilient bacterial strains. Over time, these bacteria that demonstrate antimicrobial resistance may also accumulate virulence factors driving the development of breast implant clinical infections in the first several months following breast implant placement, or in the development of high-grade capsular contractures years later (74, 75).

Furthermore, the discovery that TAPI may increase the biofilm biomass of the *S. aureus* BIAI isolates has important implications for prophylactic antibiotic treatment in breast implant-based reconstructive surgeries, as TAPI may provide the signals to promote BIAI with this pathogen. Overall, these results emphasize the need for additional studies with more clinically relevant strains to fully understand the mechanisms these pathogens encode to cause chronic BIAI. Additionally, there should be a sense of urgency in updating current protocols to include better antibiotic stewardship practices to prevent the overuse and misuse of antibiotics. Finally, future studies that dissect the pathogenic mechanisms that promote recalcitrance among BIAI are sorely needed to develop targeted therapies that can either effectively prevent or eradicate these chronic infections.

## MATERIALS AND METHODS

**Strains and growth conditions.** All strains used in this study are listed in Table 1 (51, 76). The *S. aureus* and *P. aeruginosa* isolates that caused BIAIs were provided by Margaret Olsen at Washington University in St. Louis School of Medicine. Brain-heart infusion (BHI) broth (BD, catalog no. 237200) and agar (BD, catalog no. 214010) plates were used to maintain and prepare all cultures for experiments. For the MIC and MBC assays, bacterial isolates were grown in Mueller-Hinton broth (MHB) (Sigma; catalog no. 7019-100G). For animal experiments, cells were harvested from overnight cultures grown at 37°C, shaking in BHI broth.

**MIC and MBC assays.** MIC and MBC assays were performed following the Clinical and Laboratory Standards Institute (CLSI) guidelines and as we have previously published (24, 76, 77). Briefly, *S. aureus* was grown to an optical density at 600 nm ($OD_{600}$) of 0.4, which corresponds to $\sim$2.8 $\times$ $10^8$ CFU/mL, and then diluted to $\sim$1 $\times$ $10^6$ CFU/mL. *P. aeruginosa* was grown for 4 h and diluted once to achieve $\sim$1 $\times$ $10^6$ CFU/mL. Individual antibiotics, including bacitracin (Thermo Fisher, catalog no. 226100050), cefazolin (Thermo Fisher, catalog no. 455210010), and gentamicin (Thermo Fisher, catalog no. 455310050), were diluted to concentrations ranging from 0.0625 to 2,800 $\mu$g/mL in MBH. Antibiotics were then added to each bacterial suspension at a 1:1 ratio (100 $\mu$L bacteria:100 $\mu$L antibiotic) in a 96-well plate (Fisher Scientific, catalog no. 07-000-108), resulting in a final concentration of 5 $\times$ $10^5$ CFU/mL of bacteria and antibiotics ranging from 0.03125 to 1,400 $\mu$g/mL. Controls included MHB medium alone, bacteria in MHB medium alone, and MHB with antibiotics alone. The 96-well plates were then incubated overnight at 37°C. The $OD_{600}$ was measured using a Synergy H1 Biotek microtiter plate reader, and the MICs were determined based on an $OD_{600}$ value of less than 0.1. CFU were enumerated from the MIC assays to determine the MBCs, which are based on a 99% reduction of CFU. Each experiment contained three replicates and was repeated at least twice. MIC breakpoints were obtained from the CLSI for gentamicin and cefazolin and the knowledgebase for bacitracin and are as follows: $\geq$1 $\mu$g/mL for gentamicin, $\geq$4 $\mu$g/mL for cefazolin, and $\geq$8 $\mu$g/mL for bacitracin for *S. aureus* and $\geq$16 $\mu$g/mL for gentamicin, $\geq$8 $\mu$g/mL for cefazolin, and 700 $\mu$g/mL for bacitracin for *P. aeruginosa* (76, 78). Additionally, all three antibiotics were combined into the TAPI solution used in patients: 80 mg gentamicin, 1 g cefazolin, and 50,000 U bacitracin in 500 mL saline (50). Susceptibility to the TAPI solution was measured using the same procedure described above for the MIC and MBC assays. Each experiment contained three replicates and was repeated at least twice.

**WGS of the *S. aureus* and *P. aeruginosa* BIAI isolates.** Short- and long-read sequencing methods were used to WGS all BIAI isolates as previously described (79, 80). Briefly, the *S. aureus* and *P. aeruginosa* strains were grown in BHI and LB, respectively, for 3 h, shaking at 37°C. The cultures were spun down at 10,000 $\times$ *g* for 5 min, the supernatant was discarded, and the genomic DNA was then extracted using the Qiagen QIAamp DNA minikit (Qiagen, cat catalog no. 51306) following the manufacturer's protocol, with the exception that 1 $\mu$g/mL of lysostaphin and 20 $\mu$g/mL of lysozyme was added to the initial buffer. Sequencing was performed as previously described (79). Briefly, genomic DNA underwent library preparation using the SQK-RBK004 library preparation kit and long-read sequencing via the Oxford Nanopore GridION X5 (Oxford, UK) sequencer. Additionally, short-read sequencing was performed using the Illumina NextSeq 2000 sequencer. The combination of short- and long-read sequencing data allowed us to generate complete, continuous assemblies with defined chromosomes and plasmids for characterization as clinical isolates can have highly variable accessory genomes. Bacterial genomes were then assembled as previously described (79). Briefly, raw assemblies were created with Flye version 2.7 and then underwent error correction and polishing steps using Racon version 1.4.5 and Medaka version 0.11.5. Completed genomes were analyzed for

antimicrobial gene carriage using Abricate (https://github.com/tseemann/abricate) and the CARD resistance database and virulence determinants using the VFDB (81).

**Biofilm assays.** Biofilm assays for *S. aureus* and *P. aeruginosa* were performed as previously described (29, 35). Briefly, an overnight culture of each strain, grown in BHI for *S. aureus* or LB for *P. aeruginosa*, was subcultured to an $OD_{600}$ of 0.2 and then further diluted 1:100 in fresh BHI or LB, respectively. Next, 200 $\mu$L of the subculture was added to a 96-well plate and incubated, shaking at 37°C overnight to allow biofilm to form. The next day, the supernatant culture was then carefully removed from the top of the biofilms, and 200 $\mu$L of the antibiotic solution, which was prepared as described above and used at the concentration previously determined as the MIC, twice the MIC, and four times the MIC for each strain, was added. The plate was then incubated, shaking at 37°C for 18 to 20 h. The next day, the supernatant culture was removed, and the remaining biofilm was air-dried for 30 min, washed with sterile water, stained with 0.1% crystal violet, and washed again to remove excess crystal violet; the remaining crystal violet was then solubilized with 33% acetic acid. The $OD_{600}$ was then measured using a Synergy H1 Biotek microtiter plate reader. Additionally, *S. aureus* biofilm was also formed in the presence of 20% human plasma. For these experiments, human plasma (Sigma; catalog no. P9523-5ML) was diluted to 20% in $NaHCO_3$ (Sigma; catalog no. S8875) and added to a 96-well plate, which was then incubated at 4°C overnight, as previously described (29). $NaHCO_3$-coated wells were used as a negative control. The next day, the supernate was removed, and the *S. aureus* biofilm assay was performed as described above.

**Aggregation assays.** The *P. aeruginosa* aggregation assay was performed as previously published (35), with some modifications. Briefly, overnight cultures of *P. aeruginosa* grown in LB were diluted 1:100 in fresh LB. Aggregates were then formed by growing the strains statically at 37°C for 24 h. A 1-$\mu$L sterile loop (BD, catalog no. 220215) was used to collect uniform amounts of aggregates, which were carefully suspended in 90 $\mu$L of phosphate-buffered saline (PBS) in a 96-well plate. Gentamicin was prepared as described above and added to the wells. Aggregates without antibiotics were used as controls. The plate was then incubated statically overnight at 37°C. The next day, the plate was centrifuged for 8 min at 3,500 rpm to sediment the aggregates in the wells. The supernate was removed, all remaining aggregates were processed and stained with crystal violet, and the OD was measured as described above.

**Mouse model of BIAI.** One representative BIAI isolate (117 and 157) and one reference strain (JE2 and PAO1) for each bacterial species were selected for these studies. The mouse model was performed as previously described (31, 82). Briefly, ~$10^7$ CFU of *S. aureus* and ~$10^5$ CFU of *P. aeruginosa* were prepared in 1× PBS. The mice were anesthetized via 1% to 4% isoflurane inhalation, the hair was removed from an area of the back, and the site of surgery was disinfected using chlorhexidine and 70% isopropyl alcohol. After creating a small (~1 cm) subcutaneous pocket in the haunch of the mouse, 100 $\mu$L of either sterile saline (McKesson, catalog no. 186661) or the TAPI (prepared as described above) was administered directly into the pocket via a 1-mL syringe (BD, catalog no. 309628). The implants, which were prepared by using a 6-mm Integra Miltex Standard punch to cut small, standardized pieces from the shell of a NATRELLE INSPIRA SoftTouch (reference no. SSM-755) smooth silicone breast implant, were UV-sterilized for at least 1 h before they were inserted into the subcutaneous pocket. The incision was then closed using sutures or wound clips. Each pocket was infected with the prepared bacterial inoculum via subcutaneous injection in the area where the implant was visible under the skin. Implants and tissue samples near the implants were harvested at 1, 7, and 14 dpi, and CFU were determined as previously described (29, 31, 82). Briefly, the implants were sonicated in 1× PBS for 10 min and vortexed for 1 min twice before serial dilution and plating on agar plates. Additionally, tissue samples were weighed and then homogenized using a MP-BIO FastPrep24 bead beater for 1 min twice with a 5-min rest in between before serial dilution and plating, as described above. All animal work was approved by the Animal Welfare Committee (protocol no. AWC-20-0057).

**Statistical analyses and sample size calculations.** The Mann-Whitney U test was used to determine significant differences in biomass recovered from communities formed between antibiotic treated compared to non-antibiotic-treated controls, as well as to determine significant differences between the CFU recovered from mice that received TAPI compared to those that received saline. All statistical tests were performed using GraphPad Prism 8.4.3 software.

**Data availability.** Annotated genomes were uploaded to NCBI and are available under bioproject ID numbers: PRJNA903851 (117), PRJNA903852 (158), PRJNA903854 (157), and PRJNA903855 (160).

## SUPPLEMENTAL MATERIAL

Supplemental material is available online only.
**SUPPLEMENTAL FILE 1**, PDF file, 0.2 MB.

## ACKNOWLEDGMENTS

We thank Margaret Olsen at Washington University in St. Louis School of Medicine for providing the BIAI isolates, Samantha Hitt for making the implant punches used in this study, and An Dinh and Shelby Simar for their sequencing help.

The research reported in this publication was supported by an investigator-initiated award funded by the Plastic Surgery Foundation (to J.N.W. and T.M.M.) and by funds from the University of Texas Health Science Center at Houston (to J.N.W.).

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
