## [Reviewer comments · Microbiology Spectrum]

Microbiology Spectrum

***Staphylococcus aureus* Breast Implant Infection Isolates Display Recalcitrance to Antibiotic Pocket Irrigants**

Jesus Duran Ramirez, Jana Gomez, Blake Hanson, Taha Isa, Terence Myckatyn, and Jennifer Walker

Corresponding Author(s): Jennifer Walker, University of Texas Health Science Center at Houston

Review Timeline:

Submission Date:	July 25, 2022
Editorial Decision:	October 20, 2022
Revision Received:	November 7, 2022
Accepted:	November 8, 2022

Editor: Artem Rogovskyy

Reviewer(s): Disclosure of reviewer identity is with reference to reviewer comments included in decision letter(s). The following individuals involved in review of your submission have agreed to reveal their identity: Maria Tomas (Reviewer #1)

Transaction Report:

DOI: <https://doi.org/10.1128/spectrum.02884-22>

October 20, 2022

Dr. Jennifer N Walker
University of Texas Health Science Center at Houston
Microbiology and Molecular Genetics
6431 Fannin, MSB 1.022
Houston 77030

Re: Spectrum02884-22 (*Staphylococcus aureus* Breast Implant Infection Isolates Display Recalcitrance to Antibiotic Pocket Irrigants)

Dear Dr. Jennifer N Walker:

Link Not Available

Sincerely,

Artem Rogovskyy

Journals Department
Reviewer comments:

Reviewer #1 (Comments for the Author):

The manuscript titled "Staphylococcus aureus Breast Implant Infection Isolates Display Recalcitrance to Antibiotic Pocket Irrigants" is interesting work and it is very well developed.

Only I have a major comment to consideration of the authors in relation in the experiments. The studies of the all work were developed for triplicate? Did the authors make the standard deviation of it? Please, add it the all Figures to improve the analysis. It is common to report both the standard deviation of the data and the standard error of the estimate. This point is very important to publish it in the Microbial Spectrum journal.

The conclusions of this work are appropriate and important to the clinical settings.

Reviewer #2 (Comments for the Author):

The authors of this paper have done an extensive analysis of Staph aureus and PA isolates and their response to common antibiotic cocktail used.

Some limitations include:

- There is no discussion about the proposed mechanism behind the differences noted between the staph strains. Given that the authors have access to the WGS data they can at least look at virulence factors among the strains and present that data.
- It is unclear why both long and short read sequencing was done, the authors should clarify their reasoning
- Similarly there is no discussion on what increased aggregation of a strains means clinically, and if virulence genes among these strains correlate with the phenotypes.

Staff Comments:

Preparing Revision Guidelines

Please return the manuscript within 60 days; if you cannot complete the modification within this time period, please contact me. If you do not wish to modify the manuscript and prefer to submit it to another journal, please notify me of your decision immediately so that the manuscript may be formally withdrawn from consideration by Microbiology Spectrum.

Spectrum02884-22: Review

The manuscript titled "*Staphylococcus aureus* Breast Implant Infection Isolates Display Recalcitrance to Antibiotic Pocket Irrigants" is interesting work and it is very well developed.

Only I have a major comment to consideration of the authors in relation in the experiments. The studies of the all work were developed for triplicate? Did the authors make the standard deviation of it? Please, add it the all Figures to improve the analysis. It is common to report both the standard deviation of the data and the standard error of the estimate.

This point is very important to publish it in the Microbial Spectrum journal.

The conclusions of this work are appropriate and important to the clinical settings.

We thank Dr. Artem Rogovsky for the opportunity to revise our manuscript and we would like to also thank the reviewers for their insightful comments and suggestions, which we think strengthen our manuscript. We have addressed and incorporated the reviewer's comments and concerns in the manuscript and marked them by underlining them, as well as in a point-by-point fashion below:

Response to Reviewer comments:

Reviewer #1 (Comments for the Author):

1. Only I have a major comment to consideration of the authors in relation in the experiments. The studies of the all work were developed for triplicate? Did the authors make the standard deviation of it? Please, add it the all Figures to improve the analysis. It is common to report both the standard deviation of the data and the standard error of the estimate. This point is very important to publish it in the Microbial Spectrum journal.

We thank the reviewer for bringing this unintended omission to our attention. We have added additional text within the figure legends to include the statistical test used, the p value cutoff, and the SD and SE estimates.

Reviewer #2 (Comments for the Author):

1. There is no discussion about the proposed mechanism behind the differences noted between the staph strains. Given that the authors have access to the WGS data they can at least look at virulence factors among the strains and present that data.

We thank the reviewer for this suggestion. We used the VFDB database to identify virulence determinants that might be unique to the BIAIs and added details in the Results section (lines 169-180), Discussion (lines 357-367), and the Materials and Methods (lines 456-457). Additionally, we added a Ven diagram of genes shared across all strains, those share among 2 strains, and those unique to each isolate in the supplemental materials. Using the VFDB database of known virulence factors, there were 6 genes present in BIAI strains that were absent in JE2. These genes were part of the capsule operon, cap8H, cap8I, cap8J, cap8K, and enterotoxins SEC and selK. Genes with known roles in biofilm including proteases, polysaccharide intercellular adhesin, and host matrix protein binding proteins, were encoded by all three strains.

2. It is unclear why both long and short read sequencing was done, the authors should clarify their reasoning

We thank the reviewer for bringing this to our attention. We added additional details to the materials and methods section to explain why Short and Long read methods were used. Specifically, lines: 449-452 state that we used long reads to fully and accurately assemble the genomes of the clinical isolates, which can have highly variable accessory genomes, including plasmids, that can be hard to resolve with short reads, and we used short-read sequencing to ensure the accuracy of the sequence.

3. Similarly there is no discussion on what increased aggregation of a strains means clinically, and if virulence genes among these strains correlate with the phenotypes.

We thank the reviewer for bringing this unintended omission to our attention. We have added additional clinical context to the relevance of increased bacterial community formation of particular strains clinically, and how the accumulation of virulence genes by a particular strain could manifest clinically (lines: 376-390). Here we note that breast implants can harbor bacteria in the absence of clinical sequelae, but that the accumulation of particularly virulent strains can lead to pathology.

November 8, 2022

Dr. Jennifer N Walker
University of Texas Health Science Center at Houston
Microbiology and Molecular Genetics
6431 Fannin, MSB 1.022
Houston 77030

Re: Spectrum02884-22R1 (*Staphylococcus aureus* Breast Implant Infection Isolates Display Recalcitrance to Antibiotic Pocket Irrigants)

Dear Dr. Jennifer N Walker:

Your manuscript has been accepted, and I am forwarding it to the ASM Journals Department for publication. You will be notified when your proofs are ready to be viewed.

Sincerely,

Artem Rogovskyy
Editor, Microbiology Spectrum
